# Shared regulatory networks link vein positioning and eyespot ring formation in butterflies
Tirtha Das Banerjee [1] ✉ & Antónia Monteiro [1,2] ✉

Novel traits might evolve via the repurposing of existing gene-regulatory networks. To demonstrate that a novel trait is the product of such co-option, we sought to show that the same genes and genetic interactions are taking place in different traits in a body. We examined the developmental specifications of two distinct traits that arose millions of years apart, wing veins and eyespot rings in butterflies. These two traits express a few genes in common along sharp boundaries that could derive from a shared conserved developmental program or through convergent evolution. Using laser-microdissected wing tissues followed by in-situ hybridization and antibody stainings we spatially mapped differentially expressed genes across those boundaries. We found that the expression domains of transcription factors *Optix*, *optomotor-blind*, and *spalt* in eyespot rings were similar to those observed in earlier vein positioning in butterflies and *Drosophila*. Furthermore, using CRISPR-Cas9 followed by immunostaining, we showed that *Optix* and *spalt* shared the same regulatory interaction. We propose that a primitive developmental program involved in vein positioning has been reused in the differentiation of the eyespot rings in nymphalid butterflies.

Novel organismal traits likely emerge from the expression of clusters of pre-wired genes in left contexts during development. Proposed examples of this mechanism include the reuse of a gene-regulatory network (GRN) that differentiates breathing spiracles in *Drosophila* larvae in the development of the novel posterior lobe of adult genitalia[1]; the reuse of a primitive leg GRN in patterning the novel head horns of beetles[2,3]; and the reuse of the primitive wing GRN in the development of the novel tree-hopper's helmet[4].

Butterfly eyespots are also novel traits that appear to have derived from at least two GRN co-option events. Cells at the center of these colorful wing patterns differentiate in the larval wing and express hundreds of the same genes, and use at least two of the same cis-regulatory elements, as genes that are expressed in a growing appendage[5]. Later, during the early pupal stage, these central cells signal to surrounding cells to set up the sharp rings of color in an eyespot, defined by the expression of different transcription factors[6–11]. This mechanism bears some resemblance with the mechanism that patterns the position of the veins, in earlier stages of wing development in insects, but molecular support is still very limited.

Similarities between ring differentiation and vein positioning are primarily due to the expression of two orthologous genes in similar patterns. They include the expression of *decapentaplegic* (*dpp*), a gene that codes for a small diffusible peptide, in a stripe at the anterior-posterior (AP) boundary in larval wings[8], and at the center of the eyespot pattern in early pupal wings[12]; and the presence of Spalt protein in a band straddling the *dpp* stripe in larval wings[13] and in a disc around the eyespot center cells in pupal wings[10]. However, these similarities could have emerged via convergence rather than via a shared ancestral GRN.

Here, we test whether a larger set of genes involved in venation patterning in insects, typically expressed in sharp boundaries along the position of longitudinal veins in many insect species[14–19] (Supplementary Fig. 1), could have been repurposed to differentiate the color rings of butterfly eyespots at a later stage of wing development. Some of the known vein positioning genes in *Drosophila* respond to Dpp signaling in a symmetrical fashion, such as *spalt*, while others are exclusively expressed in either the anterior or posterior compartment[20,21]. To identify such differentially expressed (DE) genes in *B. anynana*, we used laser microdissection-based transcriptomics of anterior as well as posterior larval wing compartments. Similarly, we used laser-microdissections in pupal wing tissue to identify eyespot candidate ring genes. We confirmed similarities and differences in the spatial localization of candidate genes via *in-situ* and antibody stains. Then, we tested the function of a few genes with likely roles in both vein and eyespot ring differentiation using CRISPR-Cas9.

[1]Department of Biological Sciences, National University of Singapore, Singapore, Singapore. [2]Science Division, Yale-NUS College, Singapore, Singapore.
✉e-mail: tirtha_banerjee@u.nus.edu; antonia.monteiro@nus.edu.sg

## Results

### Differential gene expression between anterior (A) and posterior (P) compartments of larval wings

To identify DE genes between the anterior and the posterior compartments of *B. anynana* larval wings, we performed DESeq2 analyses on total RNA extracted and sequenced from laser-based microdissections of each compartment. Biological replicate libraries from each compartment clustered together (Fig. 1B). Genes that were up-regulated in the anterior compartment included *aristaless (al), aristaless-like (al-like), cubitus-interruptus (ci), patched (ptc)*, and *Optix* (Fig. 1D, Supplementary Table 1), whereas posterior compartment up-regulated genes included *engrailed (en), invected (inv), hedgehog (hh), mirror*, and *araucan/caupolican* (Fig. 1D; Supplementary Table 1). Previously undescribed genes were also found to be up-regulated in the anterior compartment, such as *testis-specific gene A8 protein (A8), heat shock protein 67B1 (hsp67B1), lachesin, lebercilin*, and *down syndrome cell adhesion molecule 2 (dscam2)*, or up-regulated in the posterior compartment, such *muscle-specific protein 20 (msp20)* (Fig. 1D). The expression of these new genes were verified by HCR (Supplementary Fig. 2).

Genes expressed at the AP boundary, such as *dpp*, were not differentially expressed across anterior and posterior wing compartments. This was also observed for Dpp signaling associated genes such as the receptor *thickvein (tkv)*, and transcription factors *Mothers against decapentaplegic 6 (Mad6), Mad3*, and *Mad4* (Fig. 1D). Interestingly, *dpp* target venation genes which are specific to either anterior or posterior compartments in *Drosophila*, including *knirps* and *abrupt* [20,21], were not differentially expressed in *Bicyclus*, hinting at key differences in the venation positioning in between the two species.

### Differential gene expression between eyespot and control pupal wing tissues: Identification of the downstream targets of eyespot center signaling

To identify candidate genes for eyespot ring formation, we performed the same laser microdissection experiment in eyespots and adjacent control tissue (Fig. 1A, C) in 18–22 h pupal wings, followed by RNA-Seq and DEseq2 analysis, and HCR visualizations. Character-tree analysis showed that biological replicates from each tissue clustered together (Fig. 1C). Upregulated genes in eyespots included a Dpp signaling member *Mad6*; two known downstream targets of Dpp in *Drosophila* wing discs, *Optix*, and *optomotor-blind (omb)*, and vein specific downstream target *abrupt* [20] (Fig. 1E). In the control tissue, upregulated genes included the Dpp repressor *brinker (brk)* amongst others (Fig. 1E, Supplementary Table 2). We confirmed the spatial expression of a few of these genes using HCR in both larval wings and in 18–22 h pupal wings (Supplementary Figs. 2, 4, 7, and 9).

Below, we examine the detailed expression patterns of several candidate genes at three different stages of development. First, during the early larval stage when the vein patterning process is occurring; second, during the early pupal stage when Dpp is likely to play a role in positioning the target genes expressed in the colored ring, and third, during the late pupal stage to identify if Dpp or its likely downstream targets are still active in the eyespot center and the rings when the pigmentation process starts.

### Dpp signaling and its targets have conserved and novel expression patterns in larval wings compared to *Drosophila*

To confirm the deployment of the Dpp signaling pathway in larval wings, we first examined the spatial expression of pathway members, such as *dpp*, its receptor *tkv*, and the signal transducer *Mad*, which contains three paralogues in the *Bicyclus* genome: *Mad6, Mad3*, and *Mad4*.

In early larval wings, we observed two domains of *dpp* expression (Fig. 2A, B; Supplementary Fig. 3A, B, M, N), as previously reported [8,13], and two strong domains of *tkv* some distance away from the domains of *dpp* expression (Fig. 2E, F; Supplementary Fig. 3C, D). *Mad6* was expressed strongly in a broad domain spanning the AP boundary, where it is likely transducing the Dpp signal, and in the lower posterior compartment, where *dpp* has lower expression levels (Fig. 2C, D; Supplementary Fig. 3E, F). No expression domains for *Mad3* and *Mad4* were observed (Supplementary Fig. 3G–L).

Next, we examined the known target genes of Dpp signaling in fly wing vein differentiation, *spalt, omb*, and *Optix*, in the context of vein positioning in *Bicyclus*. The protein domains of *Optix* and *spalt* were previously reported [13], but the mRNA of these genes and that of *omb* were described in the present study in larval wings. Briefly, HCR stainings confirmed that *Optix* was expressed strongly in an anterior domain on both forewings and hindwings, consistent with the RNA-seq data (Fig. 1D, Fig. 2I, J, S, T; Supplementary Fig. 3O, P, W, X), and in an additional posterior domain on forewings (Fig. 3I, S; Supplementary Fig. 3O, W). *spalt* was expressed in four distinct domains in both hindwings and forewings (Fig. 2O, P, U, V; Supplementary Fig. 3Q, R), along with expression in the eyespot centers. *omb* was expressed along a broader domain straddling the AP boundary, as compared to *spalt* (Fig. 2G, H, M, N; Supplementary Fig. 3U, V), in the lower posterior compartment, and in the eyespot centers (Supplementary Fig. 3U, V). Furthermore, the expression domains of *Optix* and *omb* overlapped in the anterior compartment (Fig. 2K', L'), while *spalt* and *Optix* expression domains are juxtaposed along a boundary with little intermixing of cells (Fig. 2W', X'). In the lower posterior compartment, *spalt, Optix* and *omb* expression domains completely overlap each other (Fig. 2G–X, Supplementary Fig. 3S, Z).

Two genes critical in vein position in *Drosophila* are *knirps* (expressed along longitudinal vein 2, L2), and *abrupt* (L5), but we were not able to localize these genes in *Bicyclus* (Supplementary Fig. 9). *knirps* is downstream of the interaction of *spalt* and *Optix* [15,21], whereas *abrupt* is downstream of the interaction of *omb* and *brinker* [20]. These two genes were also not differentially expressed in the anterior and posterior specific transcriptome data (Fig. 1D). The absence of expression of these two genes indicates there are differences between *Drosophila* and *Bicyclus* venation patterning mechanisms.

### Dpp signaling and its targets are expressed in the eyespot field of pupal wings

Dpp signaling genes are expressed in the eyespots and are likely functioning in eyespot development. In early pupal wings, *dpp* was expressed strongly in the eyespot centers(Fig. 3A,B; Supplementary Fig. 4A), as previously reported [12], and *Mad6* was expressed strongly in the M1-Cu1 eyespot centers with weaker expression in the surrounding cells, along with a broad domain spanning the AP boundary conserved from the larval stage (Fig. 3C, Supplementary Fig. 4C). *tkv* expression was observed only in intervein cells (Fig. 3D, Supplementary Fig. 4B). Knocking out dpp by injecting synthetic single guide RNA (sgRNA) proved to be extremely difficult, likely due to its critical role in early embryonic development. Despite limited success with *dpp* CRISPR, some of the crispant showed eyespot and venation defects (Supplementary Fig. 5).

To further implicate *dpp* in eyespot size regulation, we correlated levels of its mRNA with eyespot size by evaluating previously available RNAseq data from the two seasonal forms of *B. anynana* butterflies [22]. The mRNA was extracted from dissected pupal eyespot tissue from the wet season form, as done in the current study, but also from butterflies fated to have small, dry season eyespots. Eyespot size positively correlated with levels of *dpp* (Supplementary Fig. 6A–E). Furthermore, the expression of *dpp* was restricted to the area of the future white scales, as it did not overlap with the gene *ivory*, known to be involved in the development of the surrounding black scales and co-localized with *spalt* in the eyespot center (Supplementary Fig. 6F–K).

The known targets of Dpp signaling from *Drosophila*, *spalt, Optix*, and *omb*, showed mostly conserved patterns of expression in 18–24 h pupal wings with some differences. *spalt* expression was found closest to the *dpp*-expressing cells, in the black disc, as previously shown [10], whereas *Optix* and *omb* were newly visualized co-expressed in the periphery of the eyespot, in the future orange scale cells (Fig. 3E–K; Supplementary Fig. 7A–G). Strong *omb* expression was also observed in the eyespot centers and weakly in the black scale domain overlapping *spalt* expression from 18–55 h pupal wings (Supplementary Fig. 7A–G). Taken together, the three genes formed sharp boundaries along the future white, black, and orange scale cells (Fig. 3G', K'; Supplementary Fig. 7G). The same expression patterns were observed for Optix (and Spalt) proteins using antibodies (Supplementary Fig. 8).

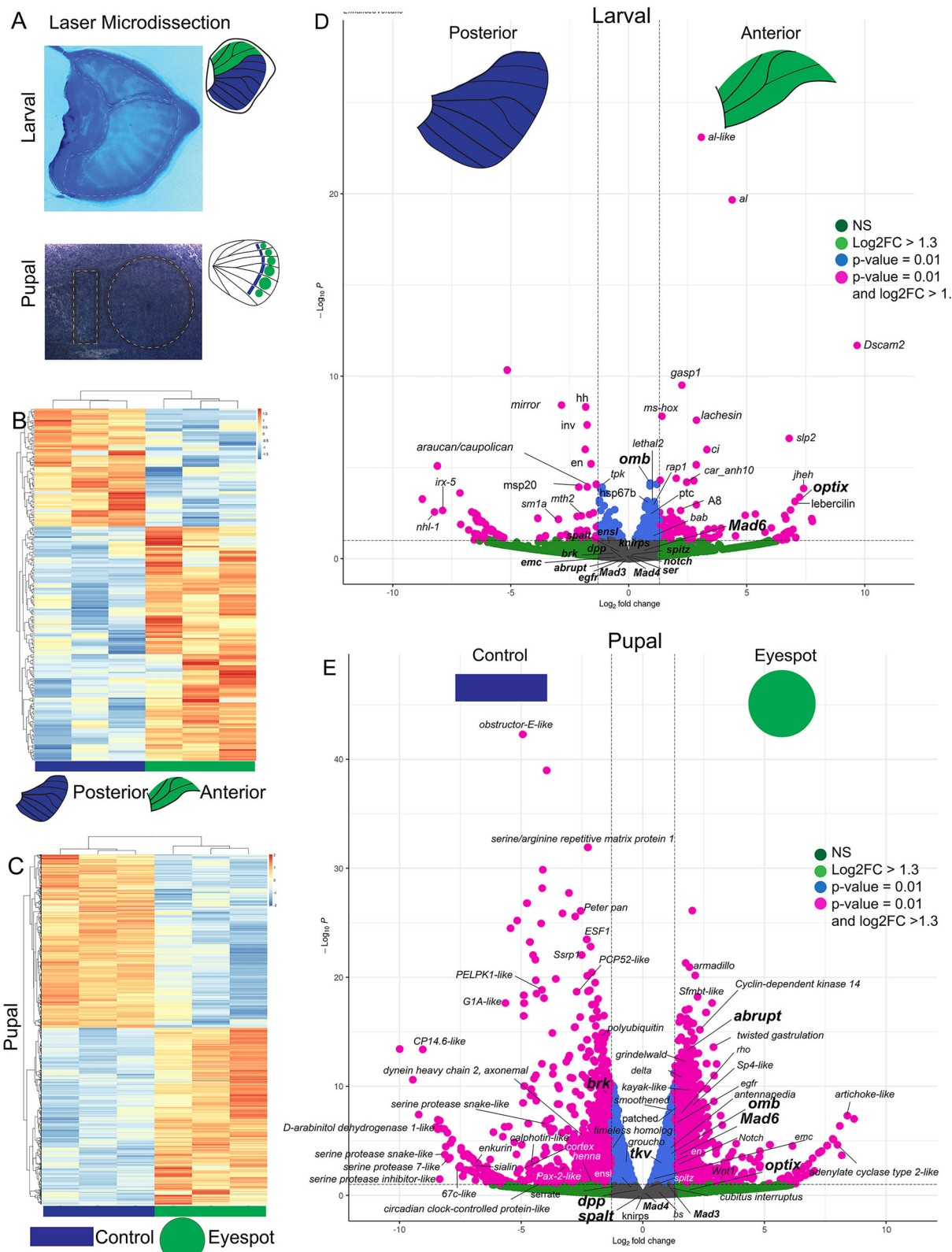

**Fig. 1 | RNA-Seq of the larval hindwing anterior and posterior compartments and of pupal hindwing eyespot and control tissues dissected with a laser. A** Laser microdissections were performed to isolate the anterior and posterior compartment of larval wing discs, and the eyespot and adjacent control tissue in 18–22 h pupal wings. **B** Heatmap clustering of the biological replicates of the anterior and the posterior compartments of larval wings and of (**C**) eyespot and adjacent control tissues of pupal wings. **D** Volcano plot showing differentially expressed genes between the anterior and the posterior compartments of larval wings and of (**E**) eyespot and control tissues of pupal wings. Threshold parameters used for the Volcano Plot: Log2FoldChange = 1.3 and $p$ = 0.01.

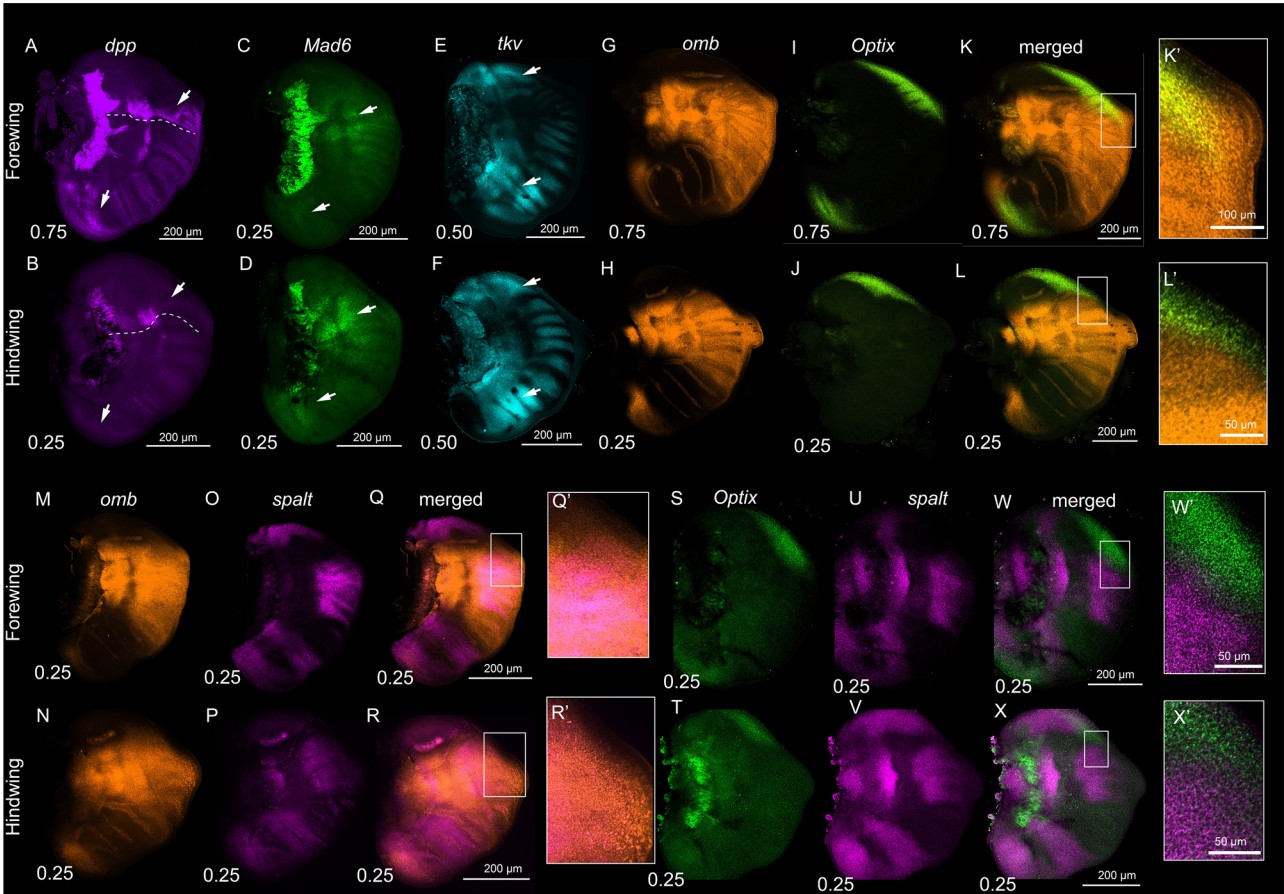

**Fig. 2 | Expression of *decapentaplegic (dpp), thickveins (tkv), Mothers against dpp 6 (Mad6), spalt, Optix,* and *omb* in the early larval wings of *B. anynana*.**
**A**, **B** Expression of *dpp* anterior to the AP compartment boundary (dotted line) in forewings and hindwings. Expression of *dpp* was also observed in the lower posterior compartment of the forewing and faintly in the hindwing (arrows) and along cells flanking the veins in forewings and hindwings. **C**, **D** Expression of *Mad6* in a broad domain spanning the AP boundary and in a small lower posterior domain (arrows), and (**E**, **F**) *tkv* in the upper anterior and middle posterior wing regions (arrows) in larval forewings and hindwings. Lower levels of *tkv* expression was also observed at lower levels in the intervein domains across the wing. **G–X, K', L', Q', R', W', X'** Expression of *omb, Optix,* and *spalt*.

From the downstream genes *knirps* and *abrupt*, only abrupt show strong expression in the eyespot centers (Supplementary Fig. 9Q, R). In the pupal transcriptomes, *knirps* was not differentially expressed, while *abrupt* was strongly upregulated in the eyespots (Fig. 1E).

### *Optix* is required for the development of orange scales, and *spalt* represses *Optix* expression from its domain

As *Optix* is newly described associated with orange scales, we tested the function of this gene using CRISPR-Cas9 in early embryos. Individuals that emerged from these injections were mosaics where some cells were disrupted for *Optix* and others were wildtype. Some crispants showed transformation of their orange-colored scales to a brown color (Fig. 4A, B; Supplementary Fig. 10), with no effect on the black scales. No visible phenotypes were observed for veins (Supplementary Fig. 10). Immunostainings targeting Optix and Spalt proteins, after CRISPR injections, showed loss of Optix proteins in the orange ring cells and intact Spalt protein in the black scale cells (Fig. 4K-Q). These results showed that *Optix* is required for the development of orange scales but is not regulating the expression domain of *spalt*. Involvement of *Optix* in the orange scale development is consistent with previous studies in *Junonia coenia* and *Vanessa Cardui*[23].

To test if Optix and Spalt have similar interactions as observed in the anterior domain of *Drosophila*[24] (Supplementary Fig. 1) we tested whether the expression of *spalt* was repressing the expression of *Optix*. *spalt* knockouts led to orange scales in the black scale region of the eyespots, as previously shown (Murugesan et al. 2022), and to severe venation defects in multiple individuals consistent with its role in vein positioning in *Drosophila* and *Bicyclus*[13,25]. Double immunostaining experiments on *spalt* crispants in pupal wings showed loss of Spalt protein, and ectopic levels of Optix protein, in the black scale disc region indicating repression of *Optix* by Spalt (Fig. 4L, O, R; Supplementary Fig. 11). Knocking out of *Optix* and *spalt* led to deletion of nucleotides mostly near the targeted sites (Fig. 4S–U; Supplementary Fig. 10K, L; Supplementary Fig. 11K). These data confirm that *spalt* represses *Optix* expression in the central black disc region of the eyespot.

### *Optix, omb,* and *spalt* continue to be expressed in late pupal wings while *dpp* and *tkv* are expressed in conserved cells around the veins

To test the involvement of Dpp signaling in older pupal wings (~55–72 h post pupation), when pigmentation starts appearing, we performed HCR on *dpp, tkv, spalt, Optix* and *omb*. We observed *dpp* expression along the vein cells and *tkv* expression, one cell thick, surrounding *dpp* (Fig. 5A–F). This pattern is consistent with previous studies on *Drosophila*, where *dpp* is involved in vein maintenance[26]. The putative targets of Dpp signaling *spalt, omb,* and *Optix* continue to be expressed in their eyespot ring domains (Fig. 5G–R; Supplementary Fig. 7H–M).

### Discussion

Our study showed that several genes implicated in positioning insect (*Drosophila*) wing veins during the larval stage show similar relative expression and regulatory interactions in a novel trait, butterfly eyespots.

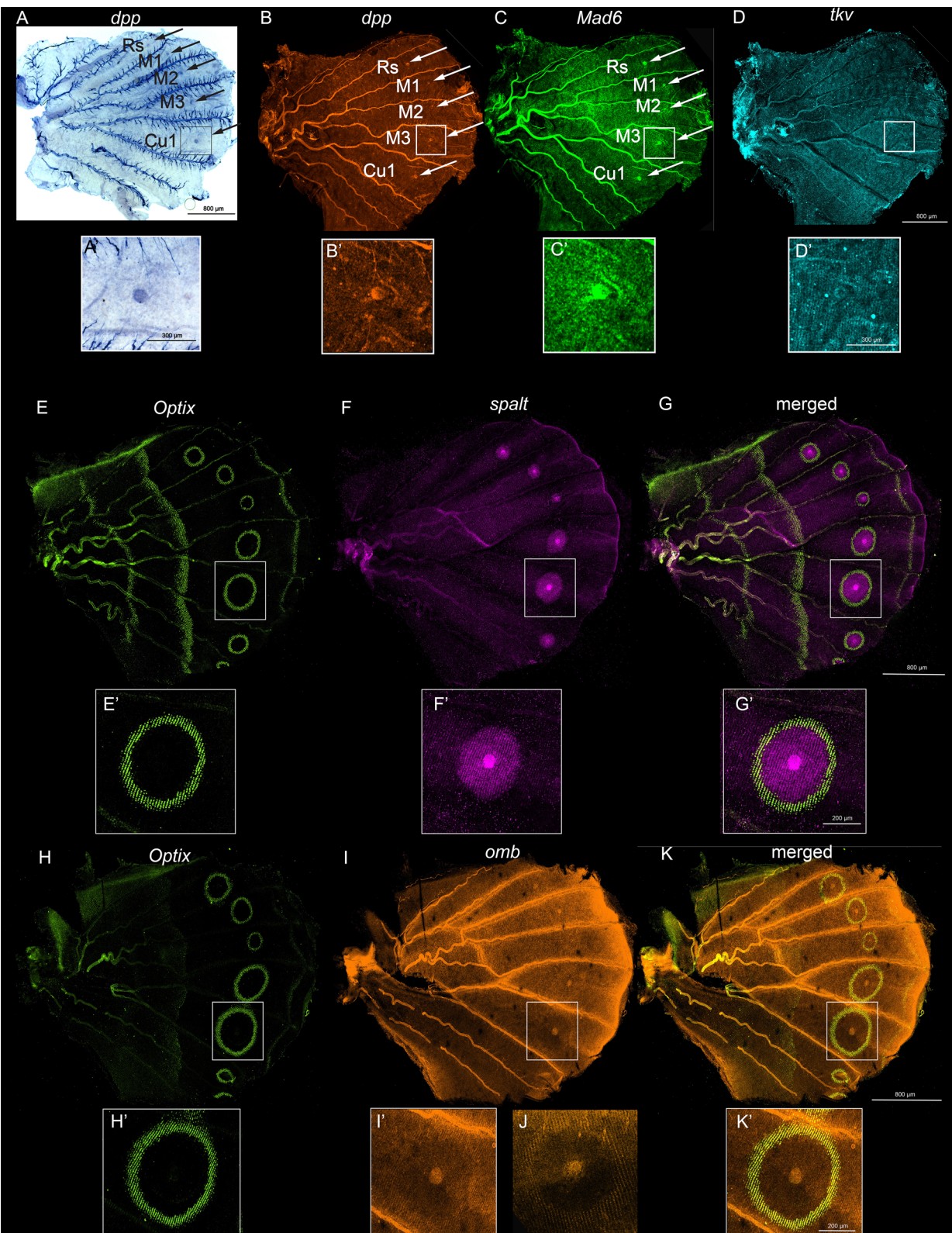

**Fig. 3 | Expression of *dpp, Mad6, tkv, Optix, spalt*, and *omb* in the 18–24 h pupal wings of *B. anynana.*** Expression of (**A**, **B**, **A'**, **B'**) *dpp* and (**C**, **C'**) *Mad6* showing expression in the Rs-Cu1 eyespot centers. *Mad6* expression was also observed in a broader domain spanning the AP boundary from the larval stage and faintly in the eyespot field. **D**, **D'** *tkv* expression was observed in the intervein cells. **E–G** and **E'–G'** Expression of *Optix* and *spalt*, and (**H–K** and **H'–K'**) *Optix* and *omb*.

**Fig. 4 | *Optix* and *spalt* are required for the development of orange and black scales, respectively, and Spalt prevents *Optix* from being expressed in the black central region of the eyespot. A**, **B** CRISPR-Cas9 knockout of *Optix* altered the development of the eyespot's orange ring in both wings (**A'**, **B'** are amplified photos of the eyespots boxed in **A**, **B**). **C**, **D** *Optix* crispants showing loss of Optix protein in cells of the eyespot orange ring. **E**, **F**, **E'**, and **F'** *spalt* crispants with orange scales developing inside the black scale domain. **G** WT eyespot, (**H**) *Optix* crispant, and (**I**) *spalt* crispant. **J**–**R** Eyespots stained with an antibody against Optix in WT, *Optix* crispant, and *spalt* crispant individuals (Note that different individuals are used for adult phenotyping and immunostains). **K** Green arrow marks the region of missing Optix protein in an *Optix* crispant that does not affect Spalt-expressing cells. **L** *spalt* crispant with Optix protein present in the eyespot central disc (pink arrow). **M**–**O** Eyespots stained with an antibody against Spalt in WT, *Optix* crispant and *spalt* crispant individuals. Blue arrow marks where Spalt proteins are missing. **P**–**R** Merged channels of Spalt and Optix. All wings were stained at 16–20 h pupal development. **S**, **T** CRISPR deletions at the *Optix* and (**U**) *spalt* target sites in three distinct individuals. The red boxes indicate the CIRISPR target sites.

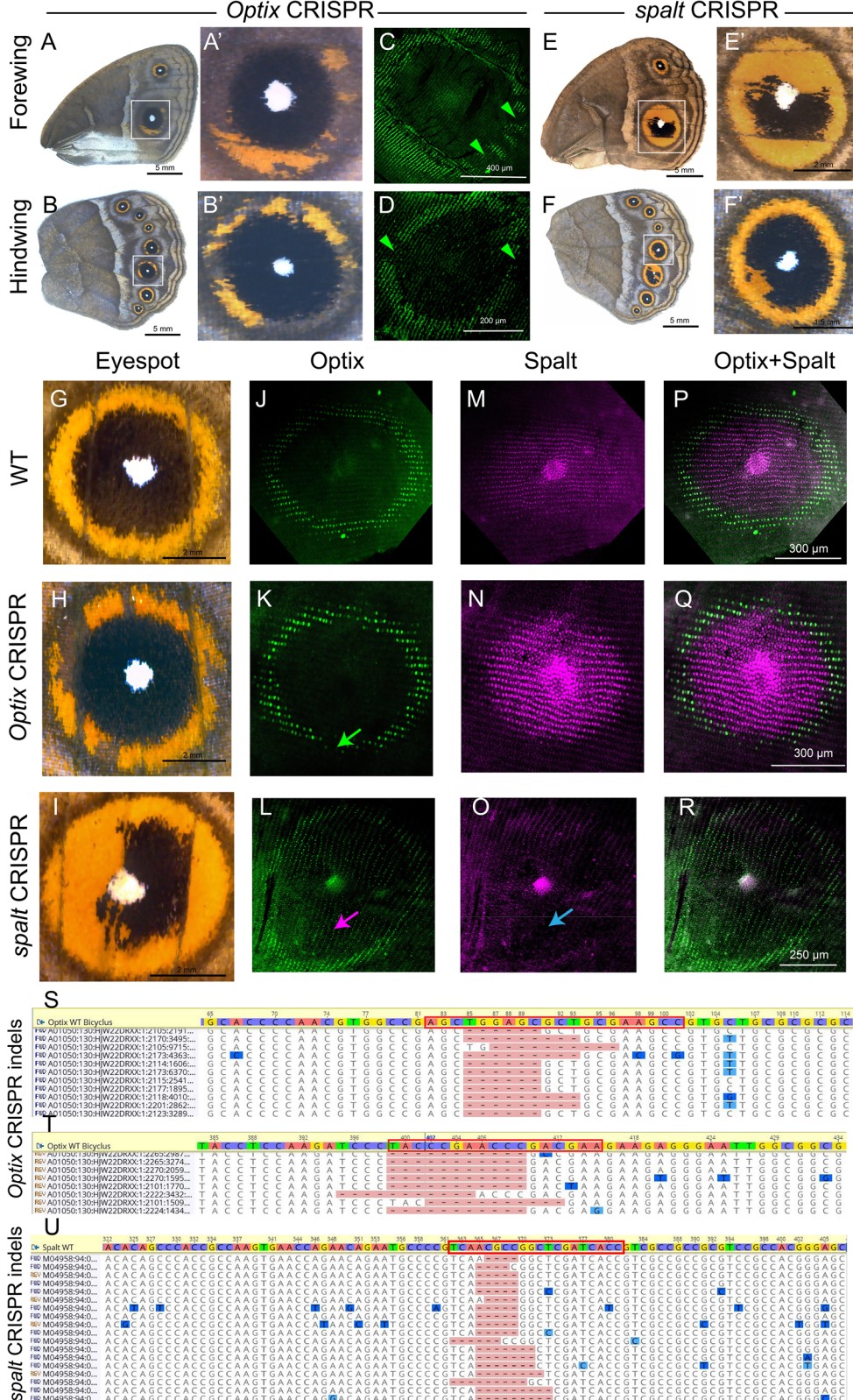

## Boundary positioning: Dpp signaling and the interaction of its target genes

We showed that Dpp mediated signaling is likely a key player in venation and in eyespot development. Its expression along the AP boundary during the larval stage, and in the eyespot centers during the pupal stage, suggests that this gene plays a key signaling role for setting up the positions of longitudinal veins in the wing, and rings in an eyespot (Fig. 6). Dpp has been implicated in the positioning and development of veins in both *Drosophila* and *Bicyclus*[13,26,27], and even though knockout data in the present study is limited to a few individuals, they showed a loss of eyespot phenotype (Supplementary Fig. 5).

There are, however, key differences in the mechanism of venation patterning in *Bicyclus* compared to *Drosophila*.

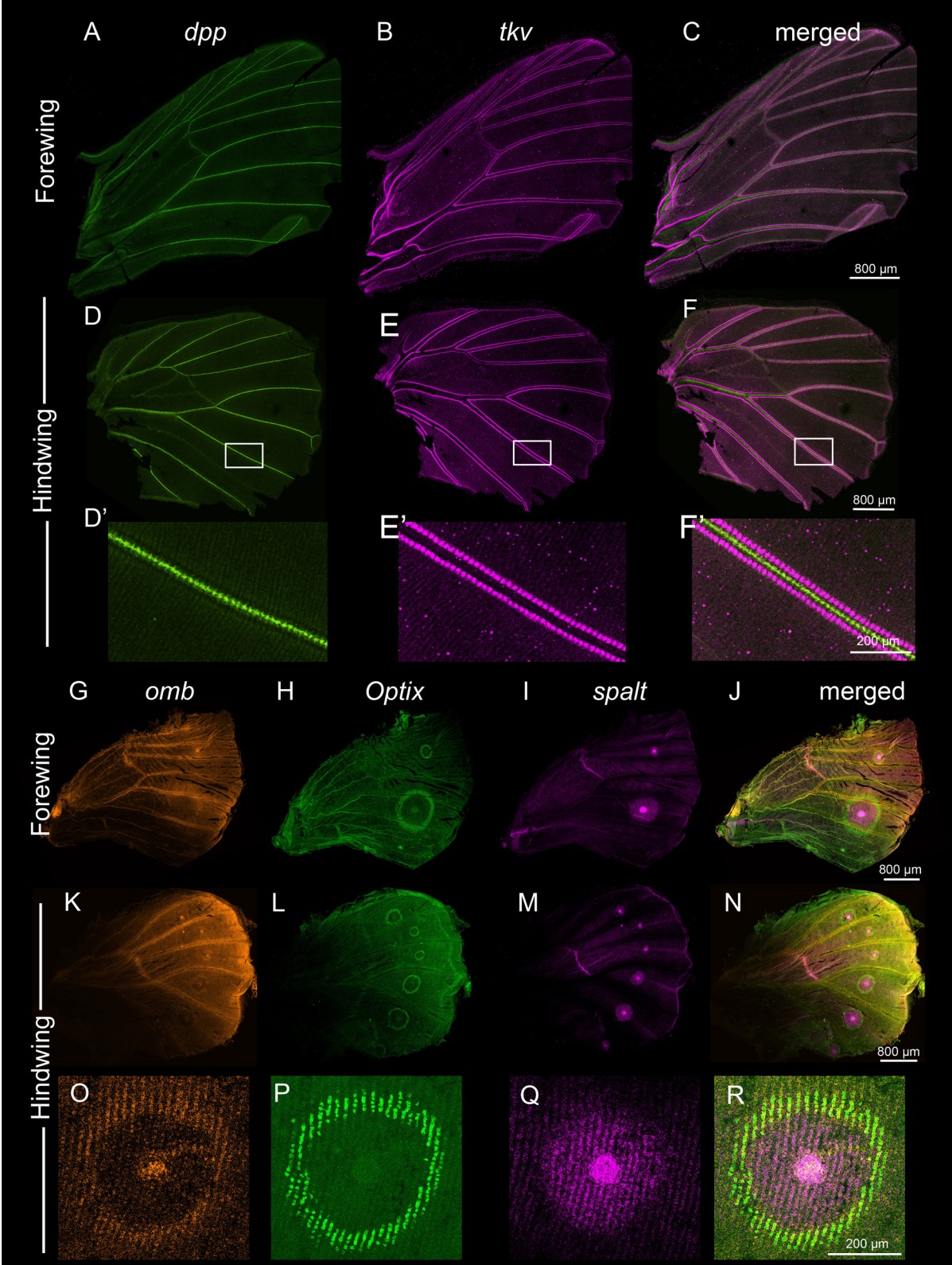

**Fig. 5 | Expression of *dpp, tkv, spalt, Optix,* and *omb* in 55–72 h pupal wings. A–F** and **D'–F'** During the later stages of pupal wing development *dpp* expression was observed restricted to the vein cells while *tkv* expressions were observed spanning the *dpp* expression domain around the veins. **G–R** *spalt*, *omb*, and *Optix* continue to express in the conserved eyespot ring domains.

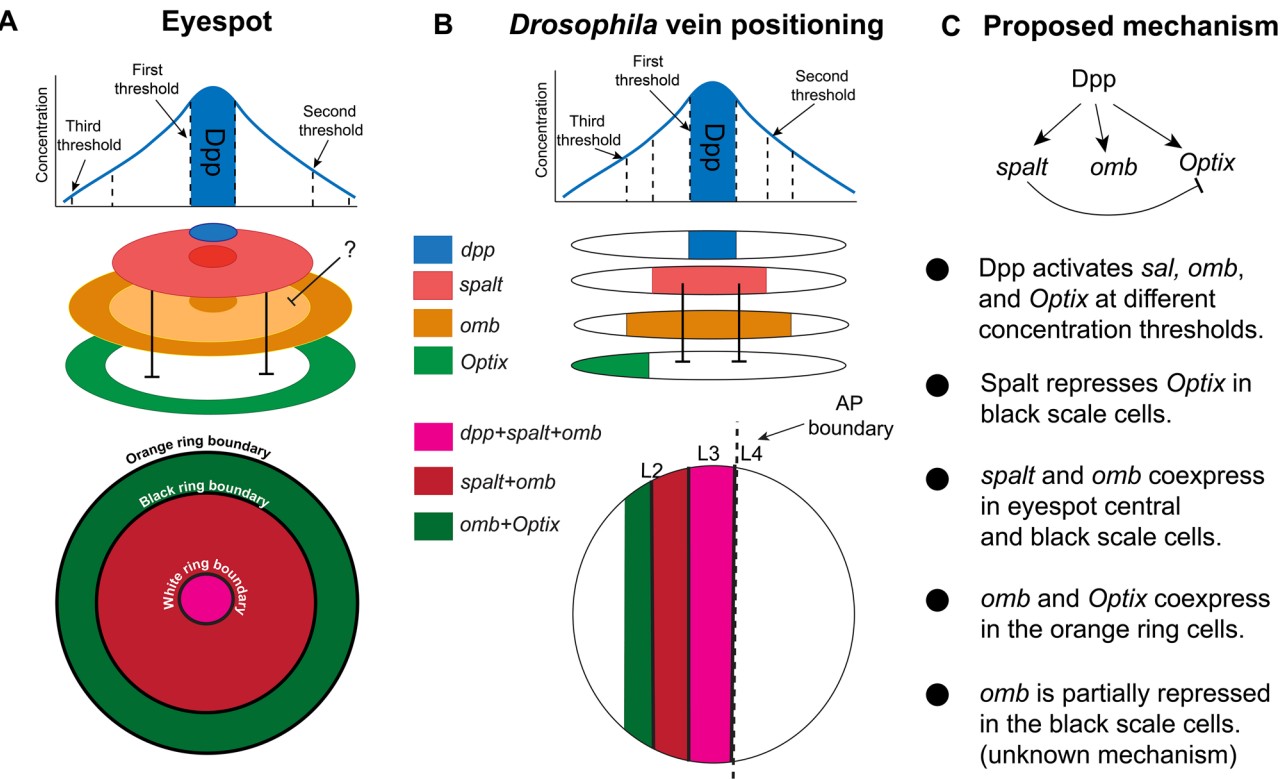

**Fig. 6 | Parallels observed in the process of setting eyespot ring boundaries in pupal wings and vein boundaries in larval wings. A** Ring boundary positioning in eyespots is carried out by a set of transcription factors. **B** These same genes are known target genes of Dpp signaling and are involved in vein positioning in *Drosophila*. **C** Proposed network of genes involved in boundary positioning in both systems.

We showed that the known downstream targets of *dpp* (in the context of vein development in *Drosophila*), *spalt*, *omb*, and *Optix*[17,24,28] were expressed in similarly broad domains spanning the central Dpp stripe, and around the eyespot centers in *Bicyclus* (Fig. 6). The activation of these genes likely involves *tkv* expressed in the intervein domain where Dpp binds and phosphorylates Mad protein (i.e., the active Mad)[29,30]. In *Bicyclus, Mad6* likely encodes the protein that transduces the Dpp signaling, as it was both expressed in a broad domain spanning the AP boundary and in the eyespot field (Fig. 3C).

We showed that *Optix* is required for orange scale differentiation, and that the negative regulatory interaction between *spalt* and *Optix* in the eyespot rings parallels that involved in L2 vein positioning in *Drosophila*[24]. We showed that *Optix* is repressed by *spalt*, leading to absence of Optix proteins in areas where Spalt proteins are found, and producing a sharp boundary between the black and orange eyespot rings (Fig. 6). In flies, a similar boundary between these two genes sets-up vein L2 in larval wing discs[24]. In *Drosophila* wing discs, down-regulation of *spalt* with RNAi results in the expansion of the Optix domain in the anterior compartment[24]. However, downregulation of *Optix* with RNAi does not affect the spatial domain of Spalt[24]. Both these results parallel our results in the eyespot rings, suggesting that a similar network is being used in venation and eyespot ring boundary development.

Additionally, we showed that *omb*, a key venation gene involved in the activation of *abrupt* along the L5 vein in *Drosophila*[20,31] is expressed in conserved domains in the larval wings of both species, and is present in the eyespot field (Fig. 6). In the *Drosophila* larval wing, *spalt* is co-expressed with *omb* at high levels of Dpp, and *Optix* is co-expressed with *omb* at lower levels of Dpp, (Fig. 6A)[20,24]. Similarly, in *Bicyclus, spalt* and *omb* are co-expressed in the eyespot center, likely receiving high levels of Dpp protein, (Fig. 6A), and *Optix* is co-expressed with *omb* in the orange ring cells, further away from the Dpp source (Fig. 6A). The target of Omb, *abrupt*, also showed strong expression in the eyespot centers (Supplementary Fig. 9Q, R). Unlike in the venation mechanism, however, lower levels of *omb* expression were found in the black scale cells, which suggest repression in that region. Aside from *spalt*, transcription factors that could be mediating such repression include *Distal-less* (*Dll*)[8,32] and newly described *A8* (Supplementary Fig. 2B). The function of *omb* in eyespot development also awaits further experiments.

### Differences in the venation patterning mechanisms of *Bicyclus* and *Drosophila*

Even though much information is available on *Drosophila* wing venation, *Drosophila* is a derived insect that might not model vein development in more basal-branching insects. For example, recent studies are proposing the involvement of reaction-diffusion mechanisms along with the classical Dpp-based positional information system in patterning the veins of moths and dragonflies[33,34]. In the present study, we observed certain key differences in *Bicyclus* relative to *Drosophila*:

(1) Loss of Optix doesn't produce venation defects in *Bicyclus* (Fig. 4, Supplementary Fig. 10)[13]. This implies that *Optix* either never had a role in basal insect venation positioning, or that this role was lost in *Bicyclus*. It is important, to note that CREs associated with Optix has been associated with venation defects[35].

(2) Lack of venation-specific genes, *knirps* and *abrupt* in the *Bicyclus* larval wings (Supplementary Fig. 9). In *Drosophila*, *spalt*, *Optix*, and *omb* are involved in positioning L2 and L5 veins via the activation of *knirps* in the L2 vein and *abrupt* in the L5 vein (Supplementary Fig. 1). Both *knirps* and *abrupt* are neither expressed nor present in the AP specific transcriptome data in the larval wings of *Bicyclus* (Fig. 1D, Supplementary Fig. 9). The absence of these genes in butterflies implies a *Drosophila*-specific recruitment for purposes of vein development, or their specific loss in butterflies.

These differences, along with multiple branchings of radial and medial veins during butterfly wing development[13] suggest that butterflies are likely using distinct mechanism of vein positioning on top of the positional-information system, which needs further exploration.

Several knowledge gaps still remain surrounding eyespot ring differentiation. Support for the vein network GRN co-option proposed in this study would benefit from testing whether *Optix, omb*, and *spalt* are responding to Dpp signals produced at the center of the eyespots, transduced through the transcription factor Mad6. Functional work associated with many of the genes whose expression patterns were discovered in the present study is yet to be carried out. It is also unclear whether Dpp and Wingless (Wnt1) are jointly required for eyespot ring differentiation. Wnt1 is another proposed morphogen for eyespot development, known to reduce eyespot size when down-regulated[11,36]. Its function should be newly investigated using knockouts. Future studies focused on the interaction between Dpp and Wnt signaling would help understand the interplay between these two pathways and their combined role in defining the eyespot rings.

In conclusion, by showing similarities[37] in the spatial expression and functionality of classic *Drosophila* venation genes, we propose that the process of differentiating insect veins, at boundaries of gene expression in the early[38] wings of flies, has been co-opted to specify the sharp rings of the eyespot pattern in later wing development in butterflies.

## Methods

### Rearing Bicyclus anynana

*Bicyclus* butterflies were raised in the lab at 27 °C, 60% humidity and 12–12 h day-night cycle. Larvae were fed with young corn leaves and adults were fed with mashed bananas.

### CRISPR-Cas9

*Optix, spalt*, and *dpp* CRISPR-Cas9 experiments were carried out based on Banerjee and Monteiro, 2018. Guides (sequence provided in Supplementary Table 3) were designed to target the coding sequence of the genes (see the supplementary file for sequences and the regions targeted). A solution containing Cas9 protein (IDT, Cat No. 1081058) and guide RNA, each at a concentration of 300 ng/ul, diluted in Cas9 buffer, and a small amount of food dye was injected into embryos. A few of these injected individuals were dissected at 24 h after pupation to perform immunostainings, while the rest were allowed to grow till adulthood. After eclosing, the adults were frozen at −20 °C and imaged under a Leica DMS1000 microscope.

For the pupal *dpp* CRISPR-Cas9 experiment, the *dpp* guide (crRNA) was mixed with an equimolar amount of tracrRNA (IDT), heated at 95 °C for 5 min and cooled to room temperature. After that Cas9 enzyme was added and left for 20 mins at room temperature. To the Cas9-guide mix, 10 µl of Cas9 plus reagent (Thermo-scientific; Cat No: CMAX00008) was added, mixed well, and another 10 µl of CRISPRMAX reagent (Thermo-scientific; Cat No: CMAX00008) was added, mixed via pipetting, and immediately injected in between the forewing and hindwing of 3–6 h old pupal wings. The pupae were left at 27 °C with 60% humidity and allowed to eclose. After eclosion, the adults were frozen and imaged under a Leica DMS1000 microscope.

For identification of the indels, DNA from the mosaic CRISPants wings were isolated using Omega Tissue DNA extraction kit (Cat No: D3396-01). Afterward PCR was performed to amplify the region of interest. PCR products were purified and sent to the next generation sequencing facility at Genewiz. After sequencing, the reads were aligned using Geneious 10.0 to the reference WT gene sequences.

### Immunostainings

Pupation time was recorded using an Olympus tough tg-6 camera, and pupal wings were dissected at different timepoints after pupation under a Zeiss Stemi 305 microscope in 1x PBS at room temperature based on a protocol previously described (Banerjee and Monteiro, 2020b). Wings were fixed using 4% formaldehyde in fix buffer (see Supplementary Table 4 for details), followed by four washes in 1x PBS, five mins each. After the washes, the wings were incubated in block buffer (see Supplementary Table 4 for details) at 4 °C overnight. The next day primary antibodies against Optix (1:3000, rat, a gift from Robert D. Reed), and Spalt (1:20000, guinea pig GP66.1), were added in wash buffer and incubated at 4 °C for 24 h. The next

day anti-rat AF488 (Invitrogen, #A-11006), anti-mouse AF488 (Invitrogen, #A28175), and anti-guinea pig AF555 (Invitrogen, # A-21435) secondary antibodies at the concentration of 1:500 in wash buffer were added followed by four washes in wash buffer, 20 min each. Wings were then mounted on an inhouse mounting media (see Supplementary Table 4 for details) and imaged under an Olympus fv3000 confocal microscope.

### Enzyme based In-situ hybridization

For the visualization of *dpp* expression in pupal wings of *Bicyclus*, pupation time was recorded, and the wings were dissected from 18–24 h old pupae in 1x PBS under a Zeiss stemi 305 microscope based on a previously described protocol (Banerjee and Monteiro, 2020b) and transferred to 1xPBST with 4% formaldehyde. After fixation for 30 min the wings were washed three times in 1x PBST for 5 mins each. The wings were then treated with proteinase K and glycine and washed again three times using 1x PBST. After, the wings were gradually transferred into pre-hybridization buffer (see Supplementary Table 5 for composition) and heated at 65 °C for 1 h. Hybridization buffer with a *dpp* probe (see Supplementary Table 5 for composition) was added to the wings and they were incubated for 16 h at 65 °C. Wings were then washed five times for 30 min each in pre-hybridization buffer. After washing, the wings were moved to room temperature and gradually transferred to 1x PBST and washed in 1x PBST. Afterwards, the wings were incubated in block buffer (see Supplementary Table 5 for composition) for 1 h, followed by addition of anti-digoxygenin (Sigma-Aldrich, Cat No. 11093274910) diluted 1/3000 times in block buffer. After 1 h of incubation the wings were washed five time, five mins each in block buffer. Finally, wings were transferred to alkaline-phosphatase buffer (see Supplementary Table S4 for composition) supplemented with NBT-BCIP (Promega, Cat No. S3771) and incubated in the dark till the development of color. The wings were imaged under a Leica DMS1000 microscope.

### Hybridization chain reaction (HCR3.0)

#### Manual methodology

HCR3.0 manually was carried out based on the protocol described in Choi et al., 2018 with the modifications for butterfly wing tissue described below. Larval and pupal wings were dissected in 1x PBS followed by fixation in 4% formaldehyde in 1x PBST. Afterwards wings were washed in 1x PBST permeabilized in a detergent solution[39] and washed with 5x SSCT. Wings were incubated for 1 h in 30% probe hybridization buffer followed by overnight incubation in 30% probe hybridization buffer supplemented with HCR probes (see supp section for sequences) at 37 °C. After incubation wings were washed five times with 30% probe wash buffer at 37 °C. Afterwards wings were brought back to room temperature and washed twice with 5x SSCT followed by incubation in amplification buffer and overnight incubation in amplification buffer supplemented with fluorescent hairpin probes (Molecular instruments) in dark for 16–20 h. The next day wings were washed four times with 5x SSCT mounted in an in-house mounting media and imaged under an Olympus fv3000 microscope.

#### Automation system based methodology

HCR3.0 automation methodology was carried out based on the protocol described in Banerjee et al., 2024. Larval and pupal wings were dissected in 1x PBS followed by fixation in 4% formaldehyde in 1x PBST. Afterwards wings were washed in 1x PBST permeabilized in a detergent or permeabilization solution[39,40] and washed with 5x SSCT. Wings were afterwards added to the microcomb plate and the script to perform the HCR3.0 reaction was executed (total reaction time 8 h). Wings were afterwards mounted in a glycerol-based media and imaged under an Olympus fv3000 microscope.

### Laser micro-dissection

Laser microdissections of the anterior and posterior compartments of larval wings and eyespot and adjacent control tissue from pupal wings were carried out based on the protocol described in ref. 41. Briefly, wings were dissected in 1x PBS and mounted in PEN membrane slides (Cat No.:

LCM0522). Wings were washed with cold 50%, 75% and 100% ethanol followed by staining using Histogene staining solution (Cat. No.: KIT0415) for 20–30 s. The wings were washed twice with cold 100% ethanol and cold acetone. The slides were either kept at −80 °C or immediately dissected using Zeiss PALM microbeam-Laser microdissection microscope using the parameters described in ref. [41]. 12 hindwings in the larval stage and 4 hindwings in the pupal stage were used for each biological replicate, respectively. A total of three biological replicates were used for downstream analysis.

### RNA-isolation from the laser dissected samples
Microsections of the larval and pupal wings were picked up from the membrane slide using a pair of fine forceps (Dumont, Cat. No. 11254-20) and transferred to 200 µl tubes with 50 µl distilled water. Afterwards, 100 µl of LCM lysis buffer[41] was added and the tubes were incubated at 55 °C in a water bath for 6 min with vortexing in between. The samples were then homogenized with 0.1 mm zirconium beads (BioSpec, Cat. No. 11079101z) in a homogenizer (Next Advance Bullet Blender) for 3 min followed by RNA isolation using a Qiagen RNA isolation kit (Cat. No. 74136). RNA samples were lyophilized in Genewiz RNA stabilization tubes and sent for sequencing (Next Generation Sequencing, Azenta US, Inc). Sequencing was performed using Novaseq PE150 system with 20 M reads per sample.

### RNA-seq analysis
Raw reads were analyzed using fastqc and cleaned using bbduk[42] with the parameters ktrim= r, k = 23, mink = 11, hdist = 1, tpe, tbo. Cleaned reads were aligned using *B. anynana* ncbi reference genome assembly (txid: 110368) using Hisat2[43] and mapped to reference *B. anynana* gtf file obtained from ncbi (txid: 110368) using stringtie[44]. A gene count matrix table was created from the stringtie files using pydr.py[44] package using python2.7 followed by analysis using DESeq2 in R[45]. Volcano plots were created using EnhancedVolcano package in R.

### Statistics
Statistical analysis in Supplementary Fig. 6 was carried out using one way analysis of variance (ANOVA: Single factor) using Microsoft Excel's Analysis ToolPak. Normalized RNAseq values from laser dissected Dry season and Wet season pupal wing eyespot data from Tian et al., 2025 were treated as independent biological replicates (four biological replicates for each seasonal form). For each gene, ANOVA tested for the significant differences in the mean expression between the seasonal forms. Statistical significance was accepted when $p < 0.05$.

### Reporting summary
Further information on research design is available in the Nature Portfolio Reporting Summary linked to this article.

### Data availability
The sequencing data generated in the paper has been submitted to NCBI under the bio-project PRJNA948390.

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

## Acknowledgements

We would like to thank Robert Reed for the anti-Optix antibody, Shen Tian and Suriya Narayanan Murugesan for helpful discussion on an earlier version of the manuscript, the DBS-CBIS confocal facility and Tong Yan for access and help with the Olympus fv3000 confocal microscope, and Tan Lu Wee for help with lab management. This work was supported by the National Research Foundation, Singapore, under its Investigatorship Program (award NRF-NRFI05-2019-0006), and Competitive Research Programme (award NRF-CRP25- 2020-0001), and MOE-Tier 1 grant (award A-8002963-00-00) from the National University of Singapore.

## Author contributions

Conceptualization: T.D.B., A.M.; Methodology and investigation: T.D.B.; Validation: T.D.B.; Formal analysis: T.D.B., A.M.; Resources: A.M.; Data curation: T.D.B.; Writing—original draft: T.D.B.; Writing—review & editing: T.D.B., A.M.; Visualization: T.D.B.; Illustration: T.D.B.; Supervision: A.M.; Project administration: A.M.; Funding acquisition: A.M.

## Competing interests

The authors declare no competing interests.
