## [Transparent Peer Review file · Communications Biology]

Setting boundaries: Butterfly eyespot rings reuse a mechanism for positioning veins

Corresponding Author: Dr Tirtha Banerjee

Version 0:

Reviewer comments:

Reviewer #1

(Remarks to the Author)

Congratulations to the authors for this excellent, already well polished article on the evo-devo on butterfly eyespots. I found the writing well calibrated to the conclusions, and the data overall of great quality and well presented.

This paper is an interesting exploration of the idea that novel traits are built from regulatory interactions that pre-exist in different contexts. The authors present compelling data with expression similarities between eyespot rings and veins, and a functional role for Spalt in these two tissues. There is a plethora of gene expression and functional analysis throughout the manuscript, all of which seem reliable and superbly put into figures. The Discussion is carefully written to nuance the conclusions, and make this manuscript particularly appreciable in my opinion.

I recommend this paper for publication without revision, and I also have the following minor suggestions to the authors for improvement.

1) 10.1016/j.cub.2024.09.073 is a recent, spectacular example of GRN co-option, could be worth citing.

2) the use of red-green channels in Figs. 4-5 makes them less accessible to scientists with dichromacy (1 man out of 12, rarer among women but I know two butterfly scientists who are among them). It should be quite easy to change the red channels to magenta.

Reviewer #2

(Remarks to the Author)

Review Comments

Thank you for giving me the opportunity to review “Reuse of an insect wing venation gene-regulatory subnetwork in patterning the eyespot rings of butterflies” by Banerjee and Monteiro. This paper presents convincing evidence that a gene regulatory network (GRN) for wing venation, ancestral to butterflies, has been repurposed to create the concentric eyespot rings found in their wings. This GRN includes the genes *dpp*, *spalt* and *optix*. This study is significant in the context of evolutionary developmental biology, as it sheds light on how conserved regulatory modules can be redeployed to generate novel morphological traits. The combination of gene expression analyses, CRISPR-based functional studies, and immunostaining makes this a valuable contribution to the field of Lepidoptera and evo-devo research.

This paper presents compelling data suggesting that an ancestral wing vein patterning gene regulatory network has been co-opted to generate the concentric ring pattern of butterfly eyespots. The study is of high scientific significance, as it sheds light on how conserved developmental modules are redeployed to produce novel morphological traits. The comprehensive approach, which encompasses gene expression analysis, functional knockouts and protein localisation, provides substantial evidence in support of the key assertions, including the pivotal role of Dpp and the conserved Spalt–Optix regulatory interaction. The argument is logical, the innovation is evident, and the conclusions are well-framed. Nevertheless, one major concern remains: the causal relationship between Dpp and the spatial domains of Spalt and Optix is still indirect.

This section could be improved, though, and I suggested below some ways of doing this that would make the paper more impactful.

Major comments

Evidence for a morphogen role of Dpp

The authors position Dpp as a “central morphogen,” proposed to be released from the eyespot center and to activate different target genes (*optix*, *spalt*) at distinct concentration thresholds. This interpretation is supported by the finding that *dpp* knockouts abolish both the orange outer ring and the black central region, indicating that Dpp contributes to the formation of both pigment rings. Furthermore, the classical gradient-threshold model is consistent with the observed expression patterns, in which *Spalt* localizes to the eyespot center while *Optix* is restricted to the surrounding ring.

However, direct evidence that Dpp indeed forms a concentration gradient is not presented in this manuscript. In particular, visualization of the spatial distribution of Dpp protein itself or a quantifiable gradient of downstream signaling activity is lacking. Overall, the current data strongly support the functional importance of *dpp*, and the morphogen hypothesis is plausible, but this portion of the argument remains somewhat indirect and would benefit from further substantiation. Demonstrating that *Spalt* and *Optix* expression is reduced or absent following *dpp* knockdown or visualizing a BMP signaling gradient across the eyespot field would substantially strengthen the argument. The reviewer understands that survival of *dpp*crispants is often poor, which makes these experiments technically challenging. Even so, partial approaches—such as lowering injection concentrations to improve viability, or presenting spatial mapping of *dpp* and/or downstream signaling activity (for example, pMad distribution) or comparing gene expression levels between the signaling center and surrounding tissue using qPCR—would already provide more direct evidence and reinforce the proposed morphogen model. In summary, the evidence presented is strong and largely supports the authors’ conclusions, but further experiments addressing gradient formation and potential contributions of additional pathways would elevate the impact and robustness of the study.

Overall, the claims and evidence are consistent and without contradiction. The novelty is clearly articulated, and the limitations are honestly acknowledged. The logical structure and reasoning are highly robust. No other significant issues arise that the reviewer deems important to highlight.

Reviewer #3

(Remarks to the Author)

Summary

The authors showed using laser-microdissection followed by RNA-seq profiling and HCR verification of genes expressed in specific regions on the wing that have been defined. This work would provide insights into how the spatial control of tissues can be harnessed to probe specific research questions, in this case the gene regulatory network underlying eyespot formation and differentiation. While many of the genes shown have validated the information in previous studies, the authors also show new genes that have not been described in eyespot patterning. These findings further support the proposition that the GRN underlying venation development has been reused for eyespot ring differentiation.

The strength of the work lies in the increased throughput to screen genes expressed at specific developmental times, which accelerates the discovery of other components of the GRN. However, it has been unclear when to draw the line between convergence and reuse, in terms of GRN use; and while authors present vein patterning and eyespot patterning as “distinct traits”, historically there has been postulations that these developmental processes might have crosstalk, so authors should proceed with careful considerations of potential confounders (e.g. veins as morphogen source and/or sink during wing development).

Major comments

Why are each of the developmental stage (larval vs 18-24h pupal vs 55-72h pupal) required to showcase the activity of gene expression across development? I suggest that to add a brief description of how each stage contributes to our overall understanding of the GRNs that are active for different aspects of the specification, determination and differentiation of the eyespot rings and/or veins. These could help guide the reader in the world of butterfly wing development.

Did the authors perform QC on RNAseq reads? Were Bioanalyzer traces obtained to verify quality of RNA before sequencing for example? I suggest to validate with RNAseq standards and provide information on read depth and sequencing technologies used, where appropriate.

The model for the GRN suggests that that first response threshold to *dpp* is flanking the expression domain of *dpp*, but I am curious why the thresholds do not overlap with the expression domain? Has there been any coexpression of *dpp* and *sal* in the same wing at 18-24h, for example, to confirm that? To what extent do *dpp* and *sal* overlap in the focus? Also, *omb* expression in the orange ring is not very obvious, I seems like *omb* is completely repressed in the black and have a low-level expression in the rest of the wing compartment, while J suggests the author’s position of *omb* expression “in the future orange scale cells”? Do the authors suggest possible differences between the different eyespots on the wing?

Minor comments

In general the data is clearly presented, however I would like to suggest 1. avoiding red-green color combinations, 2. addition of scale bars where possible and 3. clear indication of where the magnified insets are obtained from, especially when showing both whole-wing and insets in the same figure (e.g. Figs 5K-R, S6H-M, S7F-F’).

- Figure 2Q’ and R’ merged are difficult to see

- Page 5: Gene 'abrupt' instead of 'abrubt'?
- Fig 6: misspelt 'central' as 'centeral'

Version 1:

Reviewer comments:

Reviewer #2

(Remarks to the Author)

I thank the authors for their thoughtful and comprehensive revision in response to the previous round of review. The new analyses and clarifications have clearly strengthened the manuscript.

I particularly appreciate the authors' efforts to address the issue regarding the dpp morphogen hypothesis. Although the direct visualization of a Dpp gradient remains technically challenging, the authors have provided additional analyses that greatly reinforce their argument. The correlation between dpp, spalt, and optix transcript levels and eyespot size across seasonal forms offers valuable supporting evidence for the proposed model. I also appreciate the clarification of the attempts to optimize RNAi and CRISPR conditions, including the reduced guide concentrations to improve viability, which demonstrates careful experimental consideration.

Overall, the claims and evidence are coherent and mutually consistent, the novelty remains clear, and the discussion is logically structured and transparent about limitations. The revisions are thorough, scientifically, and convincingly strengthen the manuscript.

I am satisfied with the authors' responses and find the revised version ready for publication. I recommend acceptance.

Reviewer #3

(Remarks to the Author)

I agree the authors have adequately addressed my concerns and recommend for manuscript acceptance.

Below we retype the reviewer's comments in black font and our reply in blue font.

Reviewer #1 (Remarks to the Author):

Congratulations to the authors for this excellent, already well polished article on the evo-devo on butterfly eyespots. I found the writing well calibrated to the conclusions, and the data overall of great quality and well presented.

This paper is an interesting exploration of the idea that novel traits are built from regulatory interactions that pre-exist in different contexts. The authors present compelling data with expression similarities between eyespot rings and veins, and a functional role for Spalt in these two tissues. There is a plethora of gene expression and functional analysis throughout the manuscript, all of which seem reliable and superbly put into figures. The Discussion is carefully written to nuance the conclusions, and make this manuscript particularly appreciable in my opinion.

I recommend this paper for publication without revision, and I also have the following minor suggestions to the authors for improvement.

1) 10.1016/j.cub.2024.09.073 is a recent, spectacular example of GRN co-option, could be worth citing.

2) the use of red-green channels in Figs. 4-5 makes them less accessible to scientists with dichromacy (1 man out of 12, rarer among women but I know two butterfly scientists who are among them). It should be quite easy to change the red channels to magenta.

Thank you for the positive response and for calling our attention to the interesting new study on trichomes. We have decided to leave it out of the introduction, however, as this study involves detailing changes in the size of a single cell, and is somewhat different in scope from the other types of multicellular traits we are focusing on, involving the specification of multiple cell types to create a more complex trait.

We have changed the color of the red channel to magenta in our revision.

Reviewer #2 (Remarks to the Author):

Review Comments

Thank you for giving me the opportunity to review “Reuse of an insect wing venation gene-regulatory subnetwork in patterning the eyespot rings of butterflies” by Banerjee and Monteiro. This paper presents convincing evidence that a gene regulatory network (GRN) for wing venation, ancestral to butterflies, has been repurposed to create the concentric eyespot rings found in their wings. This GRN includes the genes *dpp*, *spalt* and *optix*. This study is significant in the context of evolutionary developmental biology, as it sheds light on how conserved regulatory modules can be redeployed to generate novel morphological traits. The combination of gene expression analyses, CRISPR-based functional studies, and immunostaining makes this a valuable contribution to the field of Lepidoptera and evo-devo research.

This paper presents compelling data suggesting that an ancestral wing vein patterning gene regulatory network has been co-opted to generate the concentric ring pattern of butterfly eyespots. The study is of high scientific significance, as it sheds light on how conserved developmental modules are redeployed to produce novel morphological traits. The comprehensive approach, which encompasses gene expression analysis, functional knockouts and protein localisation, provides substantial evidence in support of the key assertions, including the pivotal role of Dpp and the conserved Spalt–Optix regulatory interaction. The argument is logical, the innovation is evident, and the conclusions are well-framed. Nevertheless, one major concern remains: the causal relationship between Dpp and the spatial domains of Spalt and Optix is still indirect. This section could be improved, though, and I suggested below some ways of doing this that would make the paper more impactful.

Major comments

Evidence for a morphogen role of Dpp

The authors position Dpp as a “central morphogen,” proposed to be released from the eyespot center and to activate different target genes (*optix*, *spalt*) at distinct concentration thresholds. This interpretation is supported by the finding that *dpp* knockouts abolish both the orange outer ring and the black central region, indicating that Dpp contributes to the formation of both pigment rings. Furthermore, the classical gradient-threshold model is consistent with the observed expression patterns, in which Spalt localizes to the eyespot center while Optix is restricted to the surrounding ring.

However, direct evidence that Dpp indeed forms a concentration gradient is not presented in this manuscript. In particular, visualization of the spatial distribution of Dpp protein itself or a quantifiable gradient of downstream signaling activity is lacking.

Overall, the current data strongly support the functional importance of dpp, and the morphogen hypothesis is plausible, but this portion of the argument remains somewhat indirect and would benefit from further substantiation.

Demonstrating that Spalt and Optix expression is reduced or absent following dpp knockdown or visualizing a BMP signaling gradient across the eyespot field would substantially strengthen the argument.

The reviewer understands that survival of dpp crispants is often poor, which makes these experiments technically challenging. Even so, partial approaches—such as lowering injection concentrations to improve viability, or presenting spatial mapping of dpp and/or downstream signaling activity (for example, pMad distribution) or comparing gene expression levels between the signaling center and surrounding tissue using qPCR—would already provide more direct evidence and reinforce the proposed morphogen model.

Thank you for this comment.

It is notoriously difficult to visualize gradients of Dpp directly in tissues. One way we would be able to achieve the visualization of Dpp is by tagging it with GFP, at the endogenous locus, which is still challenging in *B. anynana*. At this stage we have not invested the time and resources required for this experiment. We have also developed our own pMad antibody but were not able to get a good signal from it.

We have, however, decided to test how Dpp mRNA levels correlate with eyespot size by evaluating previously available RNAseq data from seasonal forms of *B. anynana* butterflies. The mRNA was extracted from dissected pupal eyespot tissue from butterflies fated to have small eyespots in the dry season, and large eyespots in the wet season. Eyespot size positively correlated with levels of *dpp*, *spalt* and *Optix* mRNA. Higher levels for *spalt* and *Optix* had been shown previously by Tian et.al., 2025 (Nature Ecology and Evolution; in press), but the data for *Dpp* was added to a new supplementary figure - Figure S6.

Also, while targeting *dpp* via RNAi we did lower the concentration of the *dpp* guides to improve viability, and tried injecting embryos later in development to generate smaller edited clones of cells. None of these changes, however, improved viability.

In summary, the evidence presented is strong and largely supports the authors' conclusions, but further experiments addressing gradient formation and potential contributions of additional pathways would elevate the impact and robustness of the study. Overall, the claims and evidence are consistent and without contradiction. The novelty is clearly articulated, and the limitations are honestly acknowledged. The logical structure and reasoning are highly robust. No other significant issues arise that the reviewer deems important to highlight.

Thank you for the positive response. We agree that future work is required to test the Dpp gradient model proposed.

Reviewer #3 (Remarks to the Author):

Summary

The authors showed using laser-microdissection followed by RNA-seq profiling and HCR verification of genes expressed in specific regions on the wing that have been defined. This work would provide insights into how the spatial control of tissues can be harnessed to probe specific research questions, in this case the gene regulatory network underlying eyespot formation and differentiation. While many of the genes shown have validated the information in previous studies, the authors also show new genes that have not been described in eyespot patterning. These findings further support the proposition that the GRN underlying venation development has been reused for eyespot ring differentiation.

The strength of the work lies in the increased throughput to screen genes expressed at specific developmental times, which accelerates the discovery of other components of the GRN. However, it has been unclear when to draw the line between convergence and reuse, in terms of GRN use; and while authors present vein patterning and eyespot patterning as “distinct traits”, historically there has been postulations that these developmental processes might have crosstalk, so authors should proceed with careful considerations of potential confounders (e.g. veins as morphogen source and/or sink during wing development).

Thank you for the positive response.

Major comments

Why are each of the developmental stage (larval vs 18-24h pupal vs 55-72h pupal) required to showcase the activity of gene expression across development? I suggest that to add a brief description of how each stage contributes to our overall understanding of the GRNs that are active for different aspects of the specification, determination and differentiation of the eyespot rings and/or veins. These could help guide the reader in the world of butterfly wing development.

We have now introduced a paragraph explaining why we examined genes at different stages of development. “Below we examine the detailed expression patterns of several candidate genes at three different stages of development. First, during the early larval stage when the vein patterning process is occurring; second, during the early pupal stage when Dpp is likely to play a role in regulating the target genes expressed in the colored ring, and third, during the late pupal stage to identify if Dpp or its likely downstream targets are still active in the eyespot center and the rings when the pigmentation process starts.”

Did the authors perform QC on RNAseq reads? Were Bioanalyzer traces obtained to verify quality of RNA before sequencing for example? I suggest to validate with RNAseq standards and provide information on read depth and sequencing technologies used, where appropriate.

We outsourced the sequencing to Genewiz (Azenta). After the RNA extraction, the samples were sent to the company. Below are the data provided by the company on the samples. We added the technology and read depth in the manuscript. Sequencing was performed using Novaseq PE150 system with 20M reads per sample.

Library	Sample	Raw Reads	Raw Base(G)	Error Rate(%)	Q20(%)	Q30(%)	GC Content(%)
LA4-LFA3804_L1	LA4-LFA3804	100821843	30.247	0.04	97.91	93.99	47.69
LP4-LFA3805_L1	LP4-LFA3805	138646771	41.594	0.04	97.98	94.13	47.99
PC2-LFA3807_L1	PC2-LFA3807	39529550	11.859	0.04	98.05	94.37	49.00
PE2-LFA3806_L1	PE2-LFA3806	93741183	28.122	0.04	97.97	94.06	45.59

Library	Sample	Raw Reads	Raw Base(G)	Error Rate(%)	Q20(%)	Q30(%)	GC Content(%)
LA1-LEM10462_L1	LA1-LEM10462	25408410	7.623	0.04	97.75	93.87	48.22
LA2-LEM10463_L1	LA2-LEM10463	23439733	7.032	0.04	97.85	94.49	49.91
LP1-LEM10459_L1	LP1-LEM10459	22659223	6.798	0.04	97.96	94.22	47.46
LP2-LEM10460_L1	LP2-LEM10460	24977913	7.493	0.04	98.10	94.49	47.57
PC3-LEM10457_L1	PC3-LEM10457	24569526	7.371	0.04	97.87	94.07	48.17
PC4-LEM10458_L1	PC4-LEM10458	25646532	7.694	0.04	98.10	94.60	48.11
PE3-LEM10454_L1	PE3-LEM10454	22979335	6.894	0.04	97.85	94.07	49.10
PE4-LEM10455_L1	PE4-LEM10455	23794655	7.138	0.04	97.87	94.16	49.86

The model for the GRN suggests that that first response threshold to *dpp* is flanking the expression domain of *dpp* , but I am curious why the thresholds do not overlap with the expression domain? Has there been any coexpression of *dpp* and *sal* in the same wing at 18-24h, for example, to confirm that?

Yes, there is co-expression of *dpp* and *sal* in the eyespot center, now provided in supplementary Figure S6.

To what extent do *dpp* and *sal* overlap in the focus?

They overlap in the white center. Please refer to the newly added Figure S6F-K.

Also, *omb* expression in the orange ring is not very obvious, I' seems like *omb* is completely repressed in the black and have a low-level expression in the rest of the wing compartment, while J suggests the author's position of *omb* expression "in the future orange scale cells"? Do the authors suggest possible differences between the different eyespots on the wing?

We believe, by examining multiple stainings for *omb*, that the second pattern (in the orange ring) is the most consistent pattern across wings. *Omb* might be expressed at low levels throughout the middle sectors of a wing, presumably continuing such expression from larval stages, but in the pupal stage it is repressed in the black centers and up-regulated in the cells that will produce the orange ring. The expression in the orange ring appears to becomes stronger during the later stages of pupal wing development as shown in the manuscript figures.

Minor comments

In general the data is clearly presented, however I would like to suggest

1. avoiding red-green color combinations,

We have changed the red color to magenta.

2. addition of scale bars where possible and

We have added scale bars in the main manuscript where possible.

3. clear indication of where the magnified insets are obtained from, especially when showing both whole-wing and insets in the same figure (e.g. Figs 5K-R, S6H-M, S7F-F').

Note that in these panels the full scale eyespots are not a zoomed in version of the whole wing eyespots, they represent eyespots from other samples.

- Figure 2Q' and R' merged are difficult to see

We have adjusted the contrast on these panels.

- Page 5: Gene 'abrupt' instead of 'abrubt'?

We have corrected this misspelling.

- Fig 6: misspelt 'central' as 'centeral'

We have corrected this misspelling.

Round 2

Reviewer #2 (Remarks to the Author):

I thank the authors for their thoughtful and comprehensive revision in response to the previous round of review. The new analyses and clarifications have clearly strengthened the manuscript.

I particularly appreciate the authors' efforts to address the issue regarding the dpp morphogen hypothesis. Although the direct visualization of a Dpp gradient

remains technically challenging, the authors have provided additional analyses that greatly reinforce their argument. The correlation between *dpp*, *spalt*, and *optix* transcript levels and eyespot size across seasonal forms offers valuable supporting evidence for the proposed model. I also appreciate the clarification of the attempts to optimize RNAi and CRISPR conditions, including the reduced guide concentrations to improve viability, which demonstrates careful experimental consideration.

Overall, the claims and evidence are coherent and mutually consistent, the novelty remains clear, and the discussion is logically structured and transparent about limitations. The revisions are thorough, scientifically, and convincingly strengthen the manuscript.

I am satisfied with the authors' responses and find the revised version ready for publication. I recommend acceptance.

Thank you.